# Small RNA sequencing to differentiate lung squamous cell carcinomas from metastatic lung tumors from head and neck cancers

Yoshihisa Shimada[1]*, Jun Matsubayashi[2], Akira Saito[3], Tatsuo Ohira[1], Masahiko Kuroda[3], Norihiko Ikeda[1]

1 Department of Thoracic Surgery, Tokyo Medical University, Tokyo, Japan, 2 Department of Anatomical Pathology, Tokyo Medical University, Tokyo, Japan, 3 Department of Molecular Pathology, Tokyo Medical University, Tokyo, Japan

* zenkyu@za3.so-net.ne.jp

**Data Availability Statement:** All relevant data are within the manuscript and Supporting Information files.

## Abstract

Distinguishing lung squamous cell carcinoma (LSQCC) from a solitary metastatic lung tumor (MSQCC) from head and neck squamous cell carcinoma (HNSQCC) presents a difficult diagnostic challenge even after detailed pathological assessment. Treatment options and estimated survival outcomes after pulmonary resection differ between patients with LSQCC and MSQCC. This study aimed to investigate whether microRNA (miRNA) profiling by RNA sequencing of HNSQCC, MSQCC, and LSQCC was useful for differential diagnosis of MSQCC and LSQCC. RNA sequencing was performed to identify bioinformatically significant miRNAs from a formalin-fixed paraffin-embedded (FFPE) block from a derivation set. MiRNA levels were confirmed by validation sets using FFPE samples and serum extracellular vesicles from patients. Step-wise discriminant analysis and canonical discriminant analysis identified 13 miRNAs by which the different expression patterns of LSQCC, MSQCC, and HNSQCC groups were demonstrated. Six miRNAs (miR-10a/28/141/320b/3120) were assessed in validation sets, and 4 miRNAs (miR-10a/28/141/3120) were significantly upregulated in LSQCCs compared with MSQCCs and HNSQCCs. Serum extracellular vesicles from LSQCC patients demonstrated significantly elevated miR-10a ($p$ = .042), miR-28 ($p$ = .041), and miR-3120 ($p$ = .047) levels compared with those from MSQCC patients. RNA sequencing is useful for differential diagnosis of LSQCC and MSQCC, and the expression level of miR-10a, miR-28, and miR-3120 in serum extracellular vesicles are promising noninvasive tools for this purpose.

## Introduction

Smoking acts as a significant risk factor for lung squamous cell carcinoma (LSQCC) and head and neck squamous cell carcinoma (HNSQCC). The patient's clinical profiles and the histological appearance of both tumors are similar [1]. As it is often shown for both tumors to arise in the same patient, distinguishing an independent LSQCC from a solitary metastatic lung tumor

**Funding:** This study was supported in part by a Grant-in-Aid for Cancer Research from the Ministry of Education, Culture, Sports, and Technology, Japan (#18K08799; Y. Shimada), and internal research funding from department of thoracic surgery, Tokyo Medical University. There was no additional external funding received for this study.

**Competing interests:** The authors have declared that no competing interests exist.

**Abbreviations:** FFPE, formalin-fixed paraffin-embedded; HNSQCC, Head and neck squamous cell carcinoma; LSQCC, Lung squamous cell carcinoma; miRNA, MicroRNA; MSQCC, Metastatic tumors from head and neck squamous cell carcinoma; qRT-PCR, quantitative real-time polymerase chain reaction; SWD, step-wise discriminant.

(MSQCC) from HNSQCC presents a difficult diagnostic challenge even after detailed pathological assessment. In patients with MSQCC, a less invasive surgical procedure such as wedge lung resection is indicated as a palliative procedure, and the following systemic therapies are often available based on its pathological results and disease course. By contrast, curative-intent anatomical resection, including a segmentectomy, lobectomy, or pneumonectomy is the primary surgical procedure for LSQCC. However, those procedures have more considerable post-operative morbidities compared with wedge lung resection [2]. Treatment options and estimated survival outcomes differ between patients with LSQCC and MSQCC. Therefore, a reliable method for the differential diagnosis of these histological types before pulmonary resection is in higher demand.

MicroRNAs (miRNAs), which are endogenous and small non-coding RNAs that can function as potential oncogenes and tumor suppressors, have emerged as promising biomarkers for diagnostic purposes [3, 4]. The identification of discriminative and differentially expressed miRNA as a signature is a crucial task for cancer diagnosis and therapy [4, 5]. MiRNAs are readily detected in a formalin-fixed paraffin-embedded (FFPE) block and in various body fluids such as blood, urine, and saliva [6]. Recent reports have shown that miRNAs are transported in body fluids within extracellular vesicles [7]. Extracellular vesicles, including exosomes, are known as intercellular messengers and can shuttle cargos, such as mRNA, proteins, miRNA, and lipids between cells [8–10]. They have been studied in the cancer diagnostic setting because cancer-derived extracellular vesicles are found to be involved in metastatic cascades, such as invasion, migration, and the priming of metastatic niches [11–18]. As LSQCC and MSQCC are biologically different entities of local and metastatic disease, patient' serum-derived extracellular vesicles containing unique miRNAs can serve as noninvasive tools for their differential diagnosis.

In this study, we aimed to identify specific miRNAs for the differential diagnosis of MSQCC and LSQCC by small RNA sequencing and to quantify its miRNA levels in FFPE blocks and serum extracellular vesicles from patients with MSQCC and LSQCC for the validation analysis.

## Materials and methods

### Patients and clinical samples

LSQCC, MSQCC, and HNSQCC specimens were obtained from FFPE blocks of surgically resected tissue from patients at Tokyo Medical University Hospital. HNSQCC included patients with pharyngeal cancer, laryngeal cancer, and gingival cancer. There were a total of 28 patients who underwent surgical resection for solitary SQCC arising from the lung between January 2008 and December 2018. Of these, clinico-pathological diagnosis showed 16 patients with LSQCC and a history of HNSQCC and 12 patients with MSQCC. All 28 patients had previously undergone surgical resection for HNSQCC. For each case, pathologists reached a diagnosis of either LSQCC or MSQCC on the basis of clinical backgrounds and histologic features, including the degree of squamous differentiation and particular histological patterns (the presence of basaloid or clear cell morphology). The derivation set included 17 samples (8 LSQCCs, 6 MSQCCs, and 3 HNSQCCs) from 14 patients undergoing pulmonary resection between 2008 and 2014 that were used for small RNA sequencing (Table 1). The remaining 14 samples (8 LSQCCs and 6 MSQCCs) from 14 patients undergoing surgery between 2015 and 2018 were used for the validation study, and blood samples were also collected from the 14 patients before surgery. Each blood sample was centrifuged at 3,000 rpm for 5 minutes to separate the serum and stored at −80 <C until RNA extraction. The data of additional 41 patients (20 with stage I-II LSQCCs and 21 with stage I-III HNSQCCs) who underwent surgical resection

**Table 1. Samples from 14 patients in the derivation for small RNA sequencing.**

| No. | Type of samples | Age, sex | Primary organ | Stage of LSQCC | Type of HNSQCC | Tumor differentiation | Lung tumor location |
|-----|-----------------|----------|---------------|----------------|----------------|-----------------------|---------------------|
| 1 | LSQCC | 70, male | Lung | IB | Glottic cancer | Poor | Right upper |
| 2 | LSQCC | 60, male | Lung | IIIB | Pharyngeal cancer | Moderate | Right middle |
| 3 | LSQCC | 65, male | Lung | IA | Pharyngeal cancer | Poor | Right lower |
| 4 | LSQCC | 72, male | Lung | IIIA | Laryngeal cancer | Moderate | Right lower |
| 5 | LSQCC | 71, male | Lung | IIIB | Glottic cancer | Poor | Left upper |
| 6 | LSQCC | 76, male | Lung | IIIB | Tongue cancer | Moderate | Left upper |
| 7 | LSQCC | 75, male | Lung | IA | Laryngeal cancer | Moderate | Right upper |
| 8 | LSQCC | 80, male | Lung | IIIA | Gingival cancer | Moderate | Left upper |
|  | HNSQCC | Same as No.8 patient | | | | Moderate | - |
| 9 | MSQCC | 66, male | Pharynx | - | Pharyngeal cancer | Moderate | Left upper |
| 10 | MSQCC | 69, female | Pharynx | - | Pharyngeal cancer | Moderate | Right middle |
|  | HNSQCC | Same as No.10 patient | | | | Moderate | - |
| 11 | MSQCC | 78, female | Pharynx | - | Pharyngeal cancer | Poor | Right lower |
| 12 | MSQCC | 74, male | Larynx | - | Laryngeal cancer | Moderate | Left upper |
| 13 | MSQCC | 75, male | Tongue | - | Tongue cancer | Moderate | Left lower |
|  | HNSQCC | Same as No.13 patient | | | | Moderate | - |
| 14 | MSQCC | 74, male | Pharynx | - | Pharyngeal cancer | Poor | Right lower |

*LSQCC*, lung squamous cell carcinoma; *HNQCC*, head and neck squamous cell carcinoma; *MSQCC*, metastatic tumors from head and neck squamous cell carcinoma.

between January 2017 and December 2018 were used for the other validation study. This study was approved by the Institutional Review Board of Tokyo Medical University (study approval no. 3868). Written informed consent for the use and analyses of clinical data was obtained pre-operatively from each patient.

## Small RNA library preparation and sequencing

Total RNA of the 17 samples in the derivation set was isolated using NucleoSpin total RNA FFPE XS (Macherey-Nagel Inc. PA, USA) and was sent to Takara Bio Inc., where small RNA library preparation and sequencing were performed. Small RNA libraries were prepared using SMARTer smRNA-Seq Kit (Takara Bio Inc. Japan) for Illumina, and yields were evaluated using an Agilent 2100 BioAnalyzer. Small RNA sequencing was performed using the Illumina HiSeq 2500 system with HiSeq SBS Kit v4 reagents.

## Isolation of extracellular vesicles

Extracellular vesicles were recovered by a sequential centrifugation procedure using Exosome Isolation Kit PS (MagCapture, Fujifilm Wako. Japan). Cells were grown in T75 culture flasks for 3 to 4 days, and the conditioned media was removed from the flasks. In the case of both cell culture media and patients serum, cells were pelleted from the media by centrifugation at 300 $g$ for 5 minutes, followed by 1,200 $g$ for 20 minutes. To eliminate other cellular debris, the supernatant was spun at 10,000 $g$ for 30 minutes. The sample was concentrated by filtration (Vivaspin 20; Sartorius). After sample preparation, extracellular vesicles were purified by Mag-Capture according to the manufacturer's instructions. Extracellular vesicles were verified by electron microscopy. Extracellular vesicle size and particle numbers were analyzed using the LM10 Nanoparticle Characterization System (NanoSight, Malvern Instruments, UK). The final extracellular vesicle pellet was eluted with elution buffer.

## qRT-PCR

MiRNAs were isolated using miRNeasy FFPE Kit (QIAGEN, Netherland) for FFPE samples and Total Exosome RNA and Protein Isolation Kit (Thermo Fisher Scientific) for extracellular vesicles. cDNAs were generated using the TaqMan MicroRNA Reverse Transcription Kit (Thermo Fisher Scientific). Gene-specific TaqMan MicroRNA probes (Thermo Fisher Scientific) were utilized for quantitative analyses of miRNA transcript levels of miR-10a, miR-28, miR-99b, miR-141, miR-320b, and miR-3120. Cel-miR-39 was used as internal references. Polymerase chain reactions (PCR) were performed using StepOne (Thermo Fisher Scientific), and relative expression levels were calculated using the $2^{-\Delta\Delta CT}$ method.

## Western blotting

Extracellular vesicles were lysed in RIPA lysis buffer (Thermo Fisher Scientific) containing a protease inhibitor cocktail (Roche, Switzerland). Equal amounts of total protein were loaded onto a 4% - 20% SDS-PAGE gel and then transferred onto PVDF membranes. Membranes were blocked with 5% milk and then incubated in Tris-buffered saline with Tween 20 with a CD9 primary antibody (EXOAB-CD9A-1; SBI) overnight, followed by incubation with horseradish peroxidase-conjugated secondary antibody (Sigma-Aldrich, MO, USA). Membranes were imaged on the ChemiDoc Touch Imaging System (BIO-RAD, CA, USA).

## Statistical and bioinformatic analysis

Step-wise discriminant analysis was performed to determine potential groupings of miRNAs that maximally distinguish among LSQCC, MSQCC, and HNSQCC by using known miRNAs obtained from small RNA sequencing. Then, the canonical discriminant analysis was implemented to determine the discriminant coefficient sets 1 and 2. The absolute values of coefficients correspond to variables with more exceptional discriminant ability. The coefficient values meaning indicates how much a given miRNA contributed to a discriminant function. These analyses were carried out with the Statistical Analysis Software Package R (R Project for Statistical Computing; http://www.r-project.org). The Student *t*-test was performed to compare miRNA expression levels between 2 groups with the SPSS statistical software package (version 25.0; DDR3 RDIMM, SPSS). All tests were 2-sided, and *p*-values of less than 0.05 were considered to indicate a statistically significant difference between 2 groups.

## Results

Representative pictures of LSQCC, MSQCC, and HNSQCC in the derivation set were shown in Fig 1A. A total of 160 known miRNAs and 2,079 novel non-coding RNAs were detected by small RNA sequencing. Expression changes of known miRNA among the 3 histological groups were identified by overlapping the differential gene sets, as shown in the Venn diagram in Fig 1B. We reveal 2 representative Venn diagrams of a patient with LSQCC with a past treatment history of HNSQCC (patient 8, Fig 1C), and the other patient with MSQCC with a history of HNSQCC (patient 10, Fig 1D). Step-wise linear discriminant analysis identified 13 bioinformatically significant miRNAs, and discriminant functions were given by canonical discriminant analysis. The following formula estimated linear discriminant scores 1 and 2 obtained by the miRNA expression level of interest (fragments per kilobase of exon per million reads mapped; FPKM);

$$LD\ score = A1*X1 + A2*X2 + ————— + A13*X13$$

A: coefficients of linear discriminants, X: standardized miRNA FPKM value

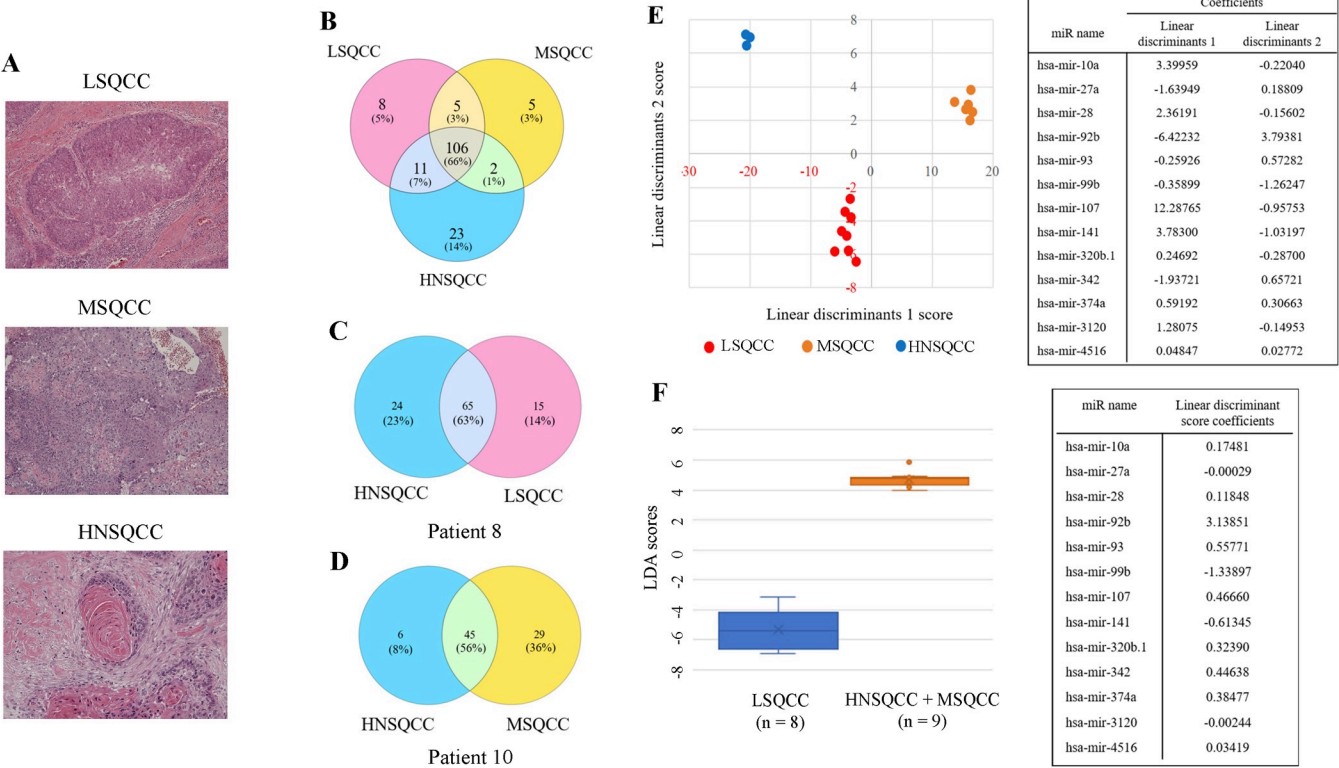

**Fig 1. MiRNA profiling by small RNA sequencing of LSQCC, MSQCC, and HNSQCC samples.** (A) Representative pathological pictures of LSQCC, MSQCC, and HNSQCC in the derivation set. (B) Venn diagram showing miRNAs that were differentially and commonly expressed in the 3 groups. (C) A diagram of a patient with LSQCC and past treatment history of HNSQCC. (D) A diagram of another patient with MSQCC and a history of HNSQCC. (E) The result of the classification with linear discriminant analysis with the list of coefficients of each miRNA's linear discriminants. Step-wise discriminant analysis of the 2 linear discriminants demonstrating that expression profiles of the 13 bioinformatically significant miRNAs can distinguish LSQCC from MSQCC and HNSQCC. (F) Linear discriminant analysis with linear discriminant score coefficients demonstrates that the expression profiles of 13 miRNAs can distinguish LSQCC from the combined group of MSQCC and HNSQCC. *LSQCC*, lung squamous cell carcinoma; *HNSQCC*, head and neck squamous cell carcinoma; *MSQCC*, metastatic tumors from HNSQCC; *LD*, linear discriminant; *miRNA*, microRNA; *FPKM*, fragments per kilobase of exon per million reads mapped.

Linear discriminants 1/ linear discriminants 2 values were on Fig 1D table. The result of the classification with linear discriminant analysis with the list of coefficients of each miRNA's linear discriminants was shown in Fig 1E. To evaluate the performance of the model with the miRNAs to distinguish LSQCC from MSQCC and HNSQCC, 1 case (LSQCC-test, MSQCC-test, and HNSQCC-test) was randomly extracted from each group as the validation case, and the rest 14 cases were used to generate the new 13-miRNA-based models. Three-fold cross-validation identified that the miRNAs were capable of differentiating LSQCC and MSQCC. Type I and II errors and an error rate obtained by the random forest algorithm in the validation sets were shown in supplementary figure legends (S1A–S1C Fig). The other linear discriminant and canonical discriminant analyses with its linear discriminant score coefficients were performed, and that made a clear distinction between LSQCC and the combined group of MSQCC and HNSQCC (Fig 1F).

Of the 13 miRNAs, 6 genes (miR-10a, miR-28, miR-99b, miR-107, miR-320b, and miR-3120) were selected for further validation analyses because the 6 miRNAs obtained from a set of linear discriminant coefficients indicated an area in proximity of the LSQCC lineage. We analyzed the quantification of the expression levels of the 6 miRNAs in 14 samples (8 LSQCC and 6 MSQCC) as an initial validation set (Fig 2). We compared the level of the 6 miRNAs in FFPE samples in 14 patients with MSQCC or LSQCC. LSQCC patients had significantly higher

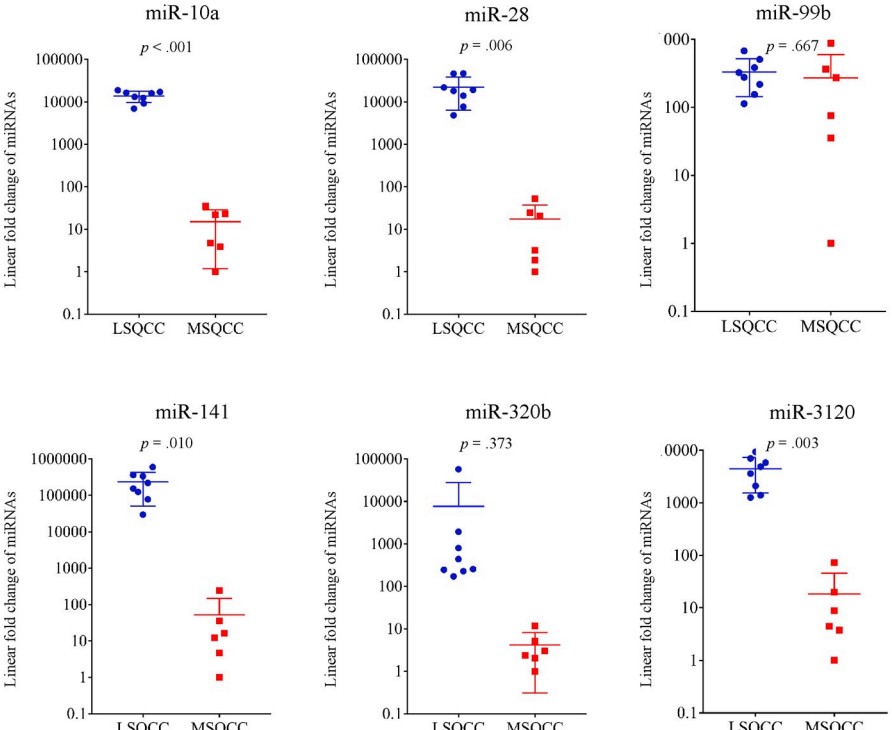

**Fig 2. The quantification of 6 miRNAs from FFPE samples in 14 patients for a validation study.** LSQCCs have significantly higher miR-10a (*p* < .001), miR-28 (*p* = .006), miR-141 (*p* = .009), and miR-3120 (*p* = .003) levels than HNSQCCs. *miRNA*, microRNA; *FFPE*, formalin-fixed paraffin-embedded; *LSQCC*, lung squamous cell carcinoma; *MSQCC*, metastatic tumors from head and neck squamous cell carcinoma.

miR-10a (*p* < .001), miR-28 (*p* = .006), miR-141 (*p* = .010), and miR-3120 (*p* = .003) levels than MSQCC patients (Fig 2). In the other validation study, the expression levels of miR-10a, miR-28, miR-141, and miR-3120 were validated between LSQCCs and HNSQCCs (Fig 3). MiR-10a (*p* < .001), miR-28 (*p* < .001), miR-141 (*p* = .002), and miR-3120 (*p* = .017) in LSQCCs are significantly higher than those in HNSQCCs (Fig 3).

Fig 4A shows extracellular vesicles that were isolated from the serum of an LSQCC patient. Extracellular vesicles were identified on transmission electron microscopy as small spherical

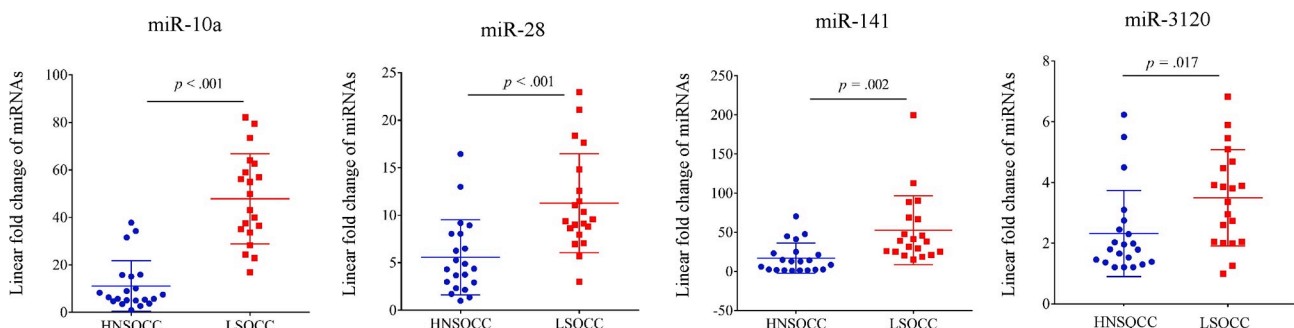

**Fig 3. Comparison of the expression of 4 miRNAs (miR-10a/28/141/3120) from FFPE block from 20 LSQCC and 21 HNSQCC patients in a validation dataset.** LSQCC samples had significantly higher miR-10a (*p* < .001), miR-28 (*p* < .001), miR-141 (*p* = .002), and miR-3120 (*p* = .017) levels than HNSQCC samples. *miRNA*, microRNA; *FFPE*, formalin-fixed paraffin-embedded; *LSQCC*, lung squamous cell carcinoma; *HNSQCC*, head and neck squamous cell carcinoma.

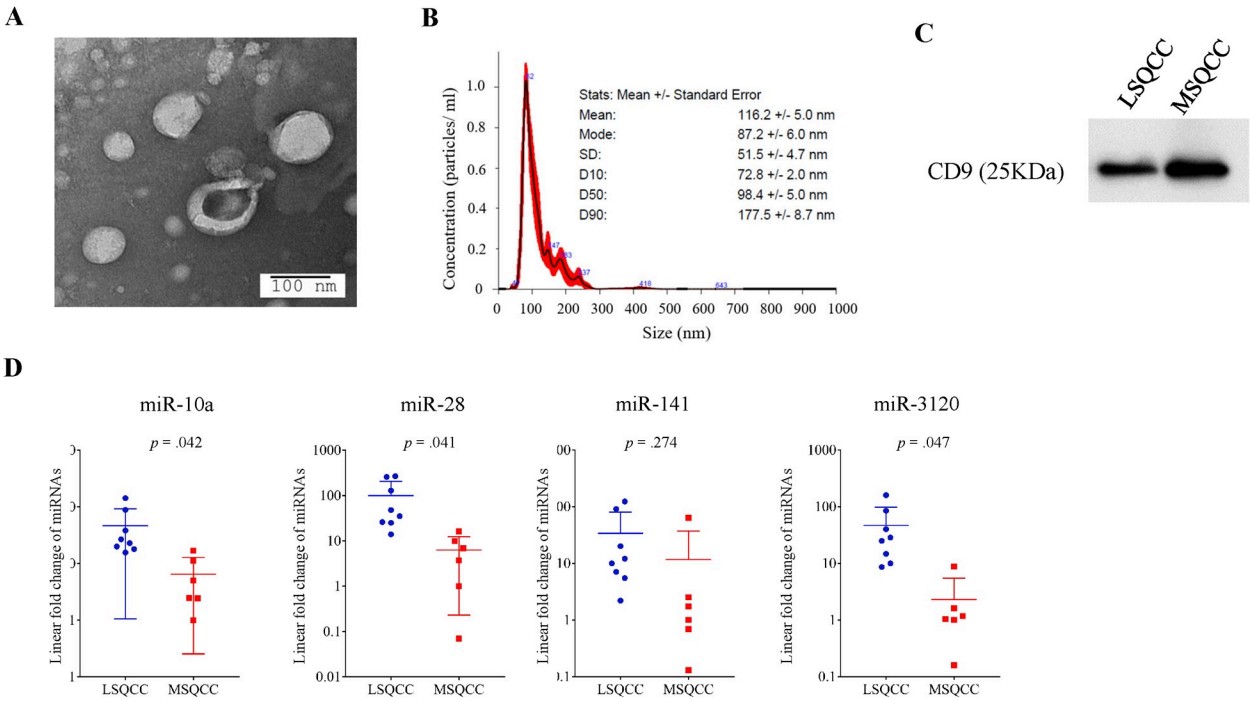

**Fig 4. Characteristics of serum extracellular vesicles from 14 patients and the quantification of selected miRNA levels in extracellular vesicles.** (A) A representative electron microscopic image of extracellular vesicles from the serum of an LSQCC patient. (B) Results of nanoparticle characterization analysis demonstrating the number of extracellular vesicles and their size distribution. (C) The protein contents of extracellular vesicles from an LSQCC and MSQCC patients were assayed by western blotting and were confirmed to express the common exosome marker CD9. (D) MiR-10a ($p$ = .042), miR-28 ($p$ = .041), and miR-3120 ($p$ = .047) levels in extracellular vesicles were significantly elevated in LSQCC patients compared with MSQCC patients. *miRNA*, microRNA; *LSQCC*, lung squamous cell carcinoma; *MSQCC*, metastatic tumors from head and neck squamous cell carcinoma.

vesicles and were characterized by the nanoparticle characterization system (Fig 4B). We checked the identification of extracellular vesicles to ensure consistency of the results in a total of 4 patients. The protein contents of extracellular vesicles were assayed by western blotting and were confirmed to express the common exosome marker CD9 (Fig 4C). MiR-10a ($p$ = .042), miR-28 ($p$ = .041), and miR-3120 ($p$ = .047) levels in extracellular vesicles in the initial validation set were significantly elevated in LSQCC patients compared with MSQCC patients (Fig 4D). S2 Fig demonstrated comparative results of the 4 miRNA levels (miR-10a/28/141/3120) in matched FFPE and serum extracellular vesicle samples.

We checked the levels of the four miRNAs using extracellular vesicles derived from 4 LSQCC cell lines (EBA-1, H520, and LK-2 purchased from ATCC) and 1 HNSQCC cell line (BHY purchased from DSMZ) to examine whether the gene expressions in patients' samples agree with those in cell lines. There were no significant differences in the expressions between the two subtypes except for miR-3120 (data not shown).

## Discussion

When a pure-solid pulmonary nodule detected in patients with a history of HNSQCC, a differential diagnosis must be made between metastasis and primary lung cancer. Primary lung cancer, including LSQCC, justifies anatomical pulmonary resection with curative intent, whereas the preferred therapeutic approach for a solitary MSQCC might be limited resection followed by systemic therapies. There have been several studies performing molecular genetic approaches to address this diagnostic issue, such as loss of heterozygosity analysis,

microsatellite analysis, p53 expression analysis, and miRNA profiling with array data using paired tumor samples from LSQCC and HNSQCC [19–22]. Geurts et al. reported that loss of heterozygosity analysis might be useful if performed together with conventional clinico-histological assessments for the differential diagnosis of the 2 types of tumors [21]. Munoz et al. demonstrated that miR-34a levels and ratios of miR-10a/miR-10b were useful in differentiating between LSQCC and MSQCC [22]. Although these may be useful additional analyses when performed together with the gold standard histological assessments, the results have been obtained only from the evaluation of surgical specimens. That indicates that those results do not affect the therapeutic decision-making process, including the selection of surgical procedures before pulmonary resection. We thus assessed whether liquid biopsy with extracellular vesicles from patient's serum was an easily-available noninvasive alternative to surgical specimens for supporting the differential diagnosis.

The validation dataset with FFPE from MSQCC and LSQCC demonstrated that miR-10a, miR-28, miR-141, and miR-3120 were significantly higher in LSQCCs than MSQCCs, while the other dataset with LSQCC and HNSQCC samples revealed that these 4 miRNAs were more abundant in LSQCCs than HNSQCCs. The expression levels of miR-10a, miR-28, and miR-3120 in serum extracellular vesicles from LSQCC patients were significantly higher than those from MSQCC patients.

MiR-10a has been reported to act as a tumor suppressor in various types of cancer, whereas a limited number of studies have reported it as an oncogene [23–26]. Munoz et al. reported that miR-10a was expressed at higher levels in LSQCC than in HNSQCC [22]. MiR-28 is known to be a tumor-suppressive miRNA in several tumor tissues and shown to be involved in the inhibition of the migration, invasion, and metastasis of a variety of cancer [27]. MiR-141 is one of the common cancer-associated miRNAs, and ovarian cancer-secreted miR-141 was reported to act as a significant mediator of intercellular communication, promoting endothelial cell angiogenesis [28]. It has been reported that miR-141 may act as a tumor suppressor in some malignancies such as gastric cancer, pancreatic cancer, and colorectal cancer [29–31]. On the other hand, miR-3120 was reported to promote stemness and invasiveness of colon cancer cells, although there have been few reports concerning the epigenetic significance of miR-3120 in cancer pathogenesis [32]. Thus, the role of miR-3120 in LSQCC and HNSQCC remains unclear. Three miRNAs, miR-10a, miR-28, and miR-141, identified by RNA sequencing in the present study, are considered tumor suppressor genes in a wide range of cancers. It can make good sense for LSQCCs as locally invasive tumors showing significantly higher expression levels of the tumor-suppressive genes compared with MSQCCs as metastatic tumors.

Extracellular vesicles, which contain a large amount of miRNA, are presently thought to be the main contributor to tumor progression and metastasis [11–13, 18, 33]. As extracellular vesicles from LSQCCs in the present study contain a significantly larger amount of miR-10a, miR-28, and miR-3120 that were confirmed in the FFPE samples compared with those from MSQCCs, these nano-size particles released from cancer cells are likely to be ideal tools for liquid biopsy to differentiate LSQCC from MSQCC or HNSQCC. However, the methods of isolation and characterization of circulating extracellular vesicles are still a matter of debate. Further studies are hence required before this liquid biopsy approach can be applied in routine clinical practice.

Although LSQCC and MSQCC samples in the derivation set from patients diagnosed based on pathological review had the potential to yield biased results, miRNA profiles of 13 miRNAs could classify all the samples into 3 groups, LSQCC, MSQCC, HNSQCC, according to the epigenetic expression. Previous studies' results were interpreted based on the premise that MSQCC and primary HNSQCC had genetically similar traits [19, 21, 22]. However, metastasis

is considered to be a clonally selective process, and HNSQCC can contain multiple clonal sub-populations capable of forming MSQCC. Therefore, our study's results support the notion that individual metastases express both different and common phenotypes, and genetic instability leads to clonal metastases becoming heterogeneous.

In conclusion, our study demonstrated that miRNA profiling by small RNA sequencing followed by validation analyses using qRT-PCR of the selected miRNAs in both FFPE samples and serum extracellular vesicles is useful for the differential diagnosis of LSQCC and MSQCC. These results led to the hypothesis that the quantification of miR-10a, miR-28, miR-141, and miR-3120 levels in not just FFPE samples but also serum extracellular vesicles may be a simple method to differentiate between HNSQCC and LSQCC lineages for the appropriate management of patients with a solitary SQCC arising in the lung before pulmonary resection even if they have a history of HNSQCC.

## Supporting information

**S1 Dataset. Minimal data set.**
(DOCX)

**S1 Fig. Three-fold cross-validation of our model with the 13 miRNAs.** Each different classification with linear discriminant analysis with the coefficients of each miRNA's linear discriminants achieved the good separation of each histologic type (A, B, and C). Type I and II errors in the validation set were 11% and 1%, respectively. An error rate obtained by the random forest algorithm was 5.88%.
(TIF)

**S2 Fig. Comparative results of the 4 miRNA levels (miR-10a/28/141/3120) in matched FFPE and serum extracellular vesicle samples.** Formalin-fixed paraffin-embedded samples from LSQCC patients had significantly higher miR-10a ($p < .001$), miR-28 ($p = .006$), miR-141 ($p = .009$), and miR-3120 ($p = .004$) levels than serum extracellular vesicle samples from the matched patients.
(TIF)

**S1 Raw image.**
(TIF)

## Acknowledgments

The authors are indebted to the medical editors from the Department of International Medical Communications of Tokyo Medical University, for editing the English manuscript. We thank Kumiko Nagase for her technical support.

## Author Contributions

**Conceptualization:** Yoshihisa Shimada, Jun Matsubayashi, Akira Saito, Tatsuo Ohira, Norihiko Ikeda.

**Data curation:** Yoshihisa Shimada, Jun Matsubayashi, Akira Saito, Tatsuo Ohira.

**Formal analysis:** Yoshihisa Shimada, Jun Matsubayashi, Akira Saito.

**Funding acquisition:** Yoshihisa Shimada.

**Investigation:** Yoshihisa Shimada, Jun Matsubayashi, Akira Saito.

**Methodology:** Yoshihisa Shimada, Jun Matsubayashi, Akira Saito.

**Software:** Yoshihisa Shimada.

**Supervision:** Tatsuo Ohira, Masahiko Kuroda, Norihiko Ikeda.

**Validation:** Yoshihisa Shimada, Masahiko Kuroda, Norihiko Ikeda.

**Writing – original draft:** Yoshihisa Shimada, Jun Matsubayashi.

**Writing – review & editing:** Yoshihisa Shimada, Jun Matsubayashi, Akira Saito, Tatsuo Ohira, Masahiko Kuroda, Norihiko Ikeda.

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
