## [Decision Letter · Decision Letter 0]

1 Oct 2020

PONE-D-20-26356

Small RNA sequencing to differentiate lung squamous cell carcinomas from metastatic lung tumors from head and neck cancers

PLOS ONE

Dear Dr. Shimada,

Thank you for submitting your manuscript to PLOS ONE. After careful consideration, we feel that it has merit but does not fully meet PLOS ONE’s publication criteria as it currently stands. Therefore, we invite you to submit a revised version of the manuscript that addresses the points raised during the review process.

We look forward to receiving your revised manuscript.

Kind regards,

Sumitra Deb, PhD

Academic Editor

PLOS ONE

Journal Requirements:

"This study was supported in part by a Grant-in-Aid for Cancer Research from the Ministry of Education, Culture, Sports, and Technology, Japan (#18K08799; Y. Shimada)."

Reviewers' comments:

Reviewer's Responses to Questions

**Comments to the Author**

1. Is the manuscript technically sound, and do the data support the conclusions?

Reviewer #1: Yes

Reviewer #2: Partly

2. Has the statistical analysis been performed appropriately and rigorously? 

Reviewer #1: Yes

Reviewer #2: No

3. Have the authors made all data underlying the findings in their manuscript fully available?

Reviewer #1: Yes

Reviewer #2: Yes

4. Is the manuscript presented in an intelligible fashion and written in standard English?

Reviewer #1: No

Reviewer #2: Yes

5. Review Comments to the Author

Reviewer #1: In this study the authors have investigated if differential levels of miRNAs in lung squamous cell carcinoma (LSQCC), solitary metastatic lung tumor (MSQCC) and head and neck squamous cell carcinoma (HNSQCC) will enable differential diagnosis of these cancer types. By using RNA sequencing, they found that LSQCC patients had significantly high levels of miR-10a, miR-28, and miR-3120 in in patient FFPE samples and were successfully able to recapitulate the same in extracellular vesicles from patient serum samples. Through this study the authors have proposed the possibility of detecting miRNA in patient extracellular vesicles (serum) as an alternative and minimally invasive approach to differentiate between HNSQCC and LSQCC patients.

Minor Revisions:

1) For consistency if data in Fig 2 and Fig 4 D is presented as a scatter dot plot as done in in Fig 3 will be more informative.

2) It is unclear if all the isolated vesicles were characterized by nanoparticle characterizing system or just some. Is Figure 4B. just a representative of sample from one patient?

3) Have the authors compared the miRNA levels in matched FFPE and serum samples? If so, it would be informative have it as supplementary data.

4) Some sections if re-written or re-phrased will improve the flow and readability of the manuscript.

a) Discussion section: Page 16, last paragraph, first two sentences read the same and have the same conclusion.

b) Discussion section: Page 18. 2nd paragraph- “miRNA profiles of 13 miRNAs could identify the 3 cancer groups as having epigenetically distinct manifestations.” It is not very clear what the author is trying say.

Reviewer #2: This study is an investigation into models for differentiating between lung squamous cell cancers and head and neck squamous cell cancers that metastasize to the lung using gene expression profiles of microRNAs and serum extracellular vesicles as a source of miRNAs.

The results are the identification of miRNAs that are of potential value for the classification based on differential expression. The use of serum extracellular vesicles is a very useful aspect.

It would help the average reader to have an explanation of how to interpret the coefficients from the discriminant analysis models.

This study shows the development and use of the models, but does not test the models for robustness or assess the potential of over-fitting. This would be done by cross-validation such as with Leave-One-Out based on the small sample size. The Type I and Type II errors and typical metrics for evaluating models, such as sensitivity and specificity, would then be presented. Results from model validation were not presented and models cannot be evaluated without this validation.

6. PLOS authors have the option to publish the peer review history of their article (what does this mean?). If published, this will include your full peer review and any attached files.

Reviewer #1: No

Reviewer #2: No

---

## [Author Response · Author response to Decision Letter 0]

20 Nov 2020

October 26, 2020

Joerg Heber

Editor-in-Chief

PLOS ONE 

Sumitra Deb

Academic editor 

PLOS ONE

Manuscript ID PONE-D-20-26356 entitled "Small RNA sequencing to differentiate lung squamous cell carcinomas from metastatic lung tumors from head and neck cancers"

Dear Dr. Heber, Deb

Thank you so much for your letter of 2-October-2020, stating your comments to our manuscript. We have carefully revised our manuscript and responded all the comments from reviewers as follows. 

Academic Editor

-COMMENT 1

Please ensure that your manuscript meets PLOS ONE's style requirements, including those for file naming. The PLOS ONE style templates can be found at https://journals.plos.org/plosone/s/file?id=wjVg/PLOSOne_formatting_sample_main_body.pdf and https://journals.plos.org/plosone/s/file?id=ba62/PLOSOne_formatting_sample_title_authors_affiliations.pdf

-ANSWER to comment 1: Thank you for this notification. We rechecked our manuscript following the style requirements. 

-COMMENT 2

Please provide additional details regarding participant consent. In the ethics statement in the Methods and online submission information, please ensure that you have specified what type you obtained (for instance, written or verbal, and if verbal, how it was documented and witnessed). If your study included minors, state whether you obtained consent from parents or guardians. If the need for consent was waived by the ethics committee, please include this information. 

-ANSWER to comment 2: Revised description is shown in Methods accordingly. 

-CHANGES: Written informed consent for the use and analyses of clinical data was obtained preoperatively from each patient. (Page 9, Line 6)

-COMMENT 3

Thank you for stating in your Funding Statement:

"This study was supported in part by a Grant-in-Aid for Cancer Research from the Ministry of Education, Culture, Sports, and Technology, Japan (#18K08799; Y. Shimada)."

-ANSWER to comment 3: According to the comment, we revised the funding statement as follows. 

-CHANGES: This study was supported in part by a Grant-in-Aid for Cancer Research from the Ministry of Education, Culture, Sports, and Technology, Japan (#18K08799; Y. Shimada), and internal research funding from department of thoracic surgery, Tokyo Medical University. There was no additional external funding received for this study. (Page 2, Line 13)

-COMMENT 4

In your Data Availability statement, you have not specified where the minimal data set underlying the results described in your manuscript can be found. PLOS defines a study's minimal data set as the underlying data used to reach the conclusions drawn in the manuscript and any additional data required to replicate the reported study findings in their entirety. All PLOS journals require that the minimal data set be made fully available. For more information about our data policy, please see http://journals.plos.org/plosone/s/data-availability. 

-ANSWER to comment 4: According to this requirement, we create a minimal data set document. 

-COMMENT 5

PLOS ONE now requires that authors provide the original uncropped and unadjusted images underlying all blot or gel results reported in a submission’s figures or Supporting Information files. This policy and the journal’s other requirements for blot/gel reporting and figure preparation are described in detail at https://journals.plos.org/plosone/s/figures#loc-blot-and-gel-reporting-requirements and https://journals.plos.org/plosone/s/figures#loc-preparing-figures-from-image-files. When you submit your revised manuscript, please ensure that your figures adhere fully to these guidelines and provide the original underlying images for all blot or gel data reported in your submission. See the following link for instructions on providing the original image data: https://journals.plos.org/plosone/s/figures#loc-original-images-for-blots-and-gels.

-ANSWER to comment 5: According to the comment, we created a figure with an uncropped image of western blotting as follows.

-CHANGES: 

-COMMENT 6

PLOS requires an ORCID iD for the corresponding author in Editorial Manager on papers submitted after December 6th, 2016. Please ensure that you have an ORCID iD and that it is validated in Editorial Manager. To do this, go to ‘Update my Information’ (in the upper left-hand corner of the main menu), and click on the Fetch/Validate link next to the ORCID field. This will take you to the ORCID site and allow you to create a new iD or authenticate a pre-existing iD in Editorial Manager. Please see the following video for instructions on linking an ORCID iD to your Editorial Manager account: https://www.youtube.com/watch?v=_xcclfuvtxQ

-ANSWER to comment 6: According to this, my ORCID iD was registered. 

Reviewer’s comments

Reviewer #1

-COMMENT 1

For consistency if data in Fig 2 and Fig 4 D is presented as a scatter dot plot as done in in Fig 3 will be more informative. 

-ANSWER to comment 1: Thank you for this advice. According to the comment, we revised figures as follows. 

-CHANGES:

Figure 4D

-COMMENT 2

It is unclear if all the isolated vesicles were characterized by nanoparticle characterizing system or just some. Is Figure 4B. just a representative of sample from one patient? 

-ANSWER to comment 2: Thank you for this comment. That is a representative of samples from one patient. We have not performed nanoparticle trafficking analysis (NTA) for all patients. As we have shown, an exosome isolation kit with membrane protein magnetic beads was used in this project, and we confirmed the identification of many small vesicles from a total of 4 patients whose size is around 100 nm in diameter in samples by NTA and electron microscope. We added the phrase as follows. 

-CHANGES: 

We checked the identification of extracellular vesicles to ensure consistency of the results in a total of 4 patients. (Page 14, Line 4)

-COMMENT 3

Have the authors compared the miRNA levels in matched FFPE and serum samples? If so, it would be informative have it as supplementary data. 

-ANSWER to comment 3: Thank you for this important advice. We can show results for comparison of gene expression from two different sources. Supplementary figure 1 is added as follows. 

-CHANGES: 

-COMMENT 4

Some sections if re-written or re-phrased will improve the flow and readability of the manuscript.

a) Discussion section: Page 16, last paragraph, first two sentences read the same and have the same conclusion.

b) Discussion section: Page 18. 2nd paragraph- “miRNA profiles of 13 miRNAs could identify the 3 cancer groups as having epigenetically distinct manifestations.” It is not very clear what the author is trying say.

-ANSWER to comment 4: Thank you for checking out these points. According to the first comment, we revised the sentence as follows. We also revised the second issue for more readable as follows. 

-CHANGES: 

a) The validation dataset with FFPE from MSQCC and LSQCC demonstrated that miR-10a, miR-28, miR-141, and miR-3120 were significantly higher in LSQCCs than MSQCCs, while the other dataset revealed that these 4 miRNAs were more abundant in LSQCCs than HNSQCCs. The expression levels of miR-10a, miR-28, and miR-3120 in serum extracellular vesicles from LSQCC patients were significantly higher than those from MSQCC patients. (Page 17, Line 3)

b) Although LSQCC and MSQCC samples in the derivation set from patients diagnosed based on pathological review had the potential to yield biased results, miRNA profiles of 13 miRNAs could classify all the samples into 3 groups, LSQCC, MSQCC, HNSQCC, according to epigenetic expression. (Page 18, Line 18)

Reviewer #2

-COMMENT 1

It would help the average reader to have an explanation of how to interpret the coefficients from the discriminant analysis models.

-ANSWER to comment 1: Thank you for this important advice. According to the comment, we added the following phrase. 

-CHANGES:

The absolute values of coefficients correspond to variables with more exceptional discriminant ability. The coefficient values meaning indicates how much a given miRNA contributed to a discriminant function. (Page 12, Line 11)

COMMENT 2

This study shows the development and use of the models, but does not test the models for robustness or assess the potential of over-fitting. This would be done by cross-validation such as with Leave-One-Out based on the small sample size. The Type I and Type II errors and typical metrics for evaluating models, such as sensitivity and specificity, would then be presented. Results from model validation were not presented and models cannot be evaluated without this validation.

-ANSWER to comment 2: Thank you for this critical point. For determining potential grouping of miRNA distinguishing three types of SQCC in this study, we performed a stepwise discriminant analysis by using known miRNAs. It might have been more understandable if we could test our discriminant model to ensure robustness and to assess the potential of over-fitting with the other different SQCC datasets or collect more SQCC cases before the initial small-RNA sequencing. However, because MSQCC is a relatively rare form of lung tumors, we could collect only 13 MSQCC cases and had to separate those samples into a training cohort and a validation cohort. Linear discriminant analysis is a method to evaluate how well a group of variables supports a priori grouping objects. We finally found significant differences of epigenetic expressions among the 3 groups by using a total of 17 SQCC samples used for small RNA-sequencing followed by the discriminant analysis (Fig 1). The model was created for just selecting the variable miRNAs set for tumor type discrimination. The selected miRNAs were tested and verified by following qRT-PCR process. Thirteen miRNAs were identified as specific variables with high discriminatory power. Because this single discriminant analysis enabled us to find out significant miRNAs for practical and mathematically analytical viewpoints, we conclude that there is no need for cross-validation of the stepwise discriminant analysis in this setting. I appreciate that the reviewer pointed it out and provided us with better opportunities to assess our model again. 

The comments offered by the reviewers and editors have been helpful in formulating what we believe is a stronger paper. We appreciate these thoughtful comments, and hope that our manuscript is now suitable for publication in PLOS ONE.

All related correspondence should be sent to Yoshihisa Shimada, M.D., Ph.D.

Department of Surgery, Tokyo Medical University Hospital

6-7-1 Nishishinjuku, Shinjyuku-ku, Tokyo, 160-0023, Japan

Phone: +81-(0)3-3342-6111, Fax: +81-(0)3-3342-6203

E-male: zenkyu@za3.so-net.ne.jp

Sincerely yours,

Yoshihisa Shimada, M.D., Ph.D.

---

## [Decision Letter · Decision Letter 1]

24 Nov 2020

PONE-D-20-26356R1

Small RNA sequencing to differentiate lung squamous cell carcinomas from metastatic lung tumors from head and neck cancers

PLOS ONE

Dear Dr. Shimada,

Thank you for submitting your manuscript to PLOS ONE. After careful consideration, we feel that it has merit but does not fully meet PLOS ONE’s publication criteria as it currently stands. Therefore, we invite you to submit a revised version of the manuscript that addresses the points raised during the review process.

We look forward to receiving your revised manuscript.

Kind regards,

Sumitra Deb, PhD

Academic Editor

PLOS ONE

Reviewers' comments:

Reviewer's Responses to Questions

**Comments to the Author**

1. If the authors have adequately addressed your comments raised in a previous round of review and you feel that this manuscript is now acceptable for publication, you may indicate that here to bypass the “Comments to the Author” section, enter your conflict of interest statement in the “Confidential to Editor” section, and submit your "Accept" recommendation.

Reviewer #1: All comments have been addressed

Reviewer #2: (No Response)

2. Is the manuscript technically sound, and do the data support the conclusions?

Reviewer #1: Yes

Reviewer #2: No

3. Has the statistical analysis been performed appropriately and rigorously? 

Reviewer #1: Yes

Reviewer #2: No

4. Have the authors made all data underlying the findings in their manuscript fully available?

Reviewer #1: Yes

Reviewer #2: Yes

5. Is the manuscript presented in an intelligible fashion and written in standard English?

Reviewer #1: Yes

Reviewer #2: Yes

6. Review Comments to the Author

Reviewer #1: (No Response)

Reviewer #2: This paper indicates that it shows differentiation between LSQCC and MSQCC using miRNAs. The title says miRNAs are used to differentiate between LSQCC and MSQCC, the Abstract and manuscript text says that the aim is to determine if miRNAs expression can be used to differentially diagnose LSQCC and MSQCC, and it’s concluded that miRNA sequencing and quantitation is useful for differential diagnosis of LSQCC and MSQCC. However, no evidence was presented that shows how well any single miRNA or combination differentiated between LSQCC and MSQCC. This problem was stated in the first review and it continues to be true in this authors’ revision. As stated in the first review, some form of metrics is required to assess if the miRNAs are capable of differentiating between LSQCC and MSQCC and thus might be useful for diagnosis. Without any assessment of how well or how poorly miRNA expression can be used to distinguish between LSQCC and MSQCC, the above-mentioned statements by the authors are incorrect. What this paper does show is the differential expression of miRNAs between LSQCC and MSQCC. If the authors restrict the paper to this topic, the paper is fine. Speculation that the identified miRNAs might be useful for distinguishing between LSQCC and MSQCC is fine.

7. PLOS authors have the option to publish the peer review history of their article (what does this mean?). If published, this will include your full peer review and any attached files.

Reviewer #1: No

Reviewer #2: No

---

## [Author Response · Author response to Decision Letter 1]

4 Dec 2020

December 5, 2020

Joerg Heber

Editor-in-Chief

PLOS ONE 

Sumitra Deb

Academic editor 

PLOS ONE

Manuscript ID PONE-D-20-26356R1 entitled "Small RNA sequencing to differentiate lung squamous cell carcinomas from metastatic lung tumors from head and neck cancers"

Dear Dr. Heber, Deb

Thank you so much for your letter of 24-November-2020, stating your comments to our manuscript. We have carefully revised our manuscript and responded all the comments from reviewers as follows. 

Reviewer #2

-COMMENT 1

This paper indicates that it shows differentiation between LSQCC and MSQCC using miRNAs. The title says miRNAs are used to differentiate between LSQCC and MSQCC, the Abstract and manuscript text says that the aim is to determine if miRNAs expression can be used to differentially diagnose LSQCC and MSQCC, and it’s concluded that miRNA sequencing and quantitation is useful for differential diagnosis of LSQCC and MSQCC. However, no evidence was presented that shows how well any single miRNA or combination differentiated between LSQCC and MSQCC. This problem was stated in the first review and it continues to be true in this authors’ revision. As stated in the first review, some form of metrics is required to assess if the miRNAs are capable of differentiating between LSQCC and MSQCC and thus might be useful for diagnosis. Without any assessment of how well or how poorly miRNA expression can be used to distinguish between LSQCC and MSQCC, the above-mentioned statements by the authors are incorrect. What this paper does show is the differential expression of miRNAs between LSQCC and MSQCC. If the authors restrict the paper to this topic, the paper is fine. Speculation that the identified miRNAs might be useful for distinguishing between LSQCC and MSQCC is fine.

-ANSWER to comment 1: Thank you for this critical comment. What the reviewer pointed out is totally true. The validation is required to assess if the miRNAs can differentiate MSQCC from LSQCC with different datasets. It is supposed to be better that the versatility of our model is evaluated with the other set of samples. However, we cannot perform the same dimensional validation because the current version of miRNA kit probe is different from the previous version of it that was used as our model. Therefore, we randomly extracted 1 case (LSQCC-test, MSQCC-test, and HNSQCC-test) each from the 3 groups as the validation case, and the rest 14 cases were used to generate the new 13-miRNA-based models. We added the following phrase and supplementary figure as follows.

-CHANGES: To evaluate the performance of the model with the miRNAs to distinguish LSQCC from MSQCC and HNSQCC, 1 case (LSQCC-test, MSQCC-test, and HNSQCC-test) was randomly extracted from each group as the validation case, and the rest 14 cases were used to generate the new 13-miRNA-based models. Three-fold cross-validation identified that the miRNAs were capable of differentiating LSQCC and MSQCC (S1 Fig A, B, C). (Page 15, Line 18)

The comments offered by the reviewers and editors have been helpful in formulating what we believe is a stronger paper. We appreciate these thoughtful comments, and hope that our manuscript is now suitable for publication in PLOS ONE.

All related correspondence should be sent to Yoshihisa Shimada, M.D., Ph.D.

Department of Surgery, Tokyo Medical University Hospital

6-7-1 Nishishinjuku, Shinjyuku-ku, Tokyo, 160-0023, Japan

Phone: +81-(0)3-3342-6111, Fax: +81-(0)3-3342-6203

E-male: zenkyu@za3.so-net.ne.jp

Sincerely yours,

Yoshihisa Shimada, M.D., Ph.D.

---

## [Decision Letter · Decision Letter 2]

8 Dec 2020

PONE-D-20-26356R2

Small RNA sequencing to differentiate lung squamous cell carcinomas from metastatic lung tumors from head and neck cancers

PLOS ONE

Dear Dr. Shimada,

Thank you for submitting your manuscript to PLOS ONE. After careful consideration, we feel that it has merit but does not fully meet PLOS ONE’s publication criteria as it currently stands. Therefore, we invite you to submit a revised version of the manuscript that addresses the points raised during the review process.

We look forward to receiving your revised manuscript.

Kind regards,

Sumitra Deb, PhD

Academic Editor

PLOS ONE

Reviewers' comments:

Reviewer's Responses to Questions

**Comments to the Author**

1. If the authors have adequately addressed your comments raised in a previous round of review and you feel that this manuscript is now acceptable for publication, you may indicate that here to bypass the “Comments to the Author” section, enter your conflict of interest statement in the “Confidential to Editor” section, and submit your "Accept" recommendation.

Reviewer #2: (No Response)

2. Is the manuscript technically sound, and do the data support the conclusions?

Reviewer #2: Yes

3. Has the statistical analysis been performed appropriately and rigorously? 

Reviewer #2: No

4. Have the authors made all data underlying the findings in their manuscript fully available?

Reviewer #2: Yes

5. Is the manuscript presented in an intelligible fashion and written in standard English?

Reviewer #2: Yes

6. Review Comments to the Author

Reviewer #2: The cross-validation is limited. All samples must be validated to avoid bias.

Readers should not have to "eyeball" the results of the validation to ascertain how well the model performed.

The performance of the model should be quantitated by Type I and Type II errors.

7. PLOS authors have the option to publish the peer review history of their article (what does this mean?). If published, this will include your full peer review and any attached files.

Reviewer #2: No

---

## [Author Response · Author response to Decision Letter 2]

15 Dec 2020

December 15, 2020

Joerg Heber

Editor-in-Chief

PLOS ONE 

Sumitra Deb

Academic editor 

PLOS ONE

Manuscript ID PONE-D-20-26356R2 entitled "Small RNA sequencing to differentiate lung squamous cell carcinomas from metastatic lung tumors from head and neck cancers"

Dear Dr. Heber, Deb

Thank you so much for your letter of 8-December-2020, stating your comments to our manuscript. We have carefully revised our manuscript and responded all the comments from reviewers as follows. 

Reviewer #2

-COMMENT 1

The cross-validation is limited. All samples must be validated to avoid bias.

Readers should not have to "eyeball" the results of the validation to ascertain how well the model performed. The performance of the model should be quantitated by Type I and Type II errors.

-ANSWER to comment 1: Thank you for this comment. Because the validation would deviate from the main line of the study, we described it in supplementary figure legends as follows.

-CHANGES: S1 Fig. Three-fold cross-validation of our model with the 13 miRNAs. Each different classification with linear discriminant analysis with the coefficients of each miRNA’s linear discriminants achieved the good separation of each histologic type (A, B, and C). Type I and II errors in the validation set were 11% and 1%, respectively. An error rate obtained by the random forest algorithm was 5.88%.

The comments offered by the reviewers and editors have been helpful in formulating what we believe is a stronger paper. We appreciate these thoughtful comments, and hope that our manuscript is now suitable for publication in PLOS ONE.

All related correspondence should be sent to Yoshihisa Shimada, M.D., Ph.D.

Department of Surgery, Tokyo Medical University Hospital

6-7-1 Nishishinjuku, Shinjyuku-ku, Tokyo, 160-0023, Japan

Phone: +81-(0)3-3342-6111, Fax: +81-(0)3-3342-6203

E-male: zenkyu@za3.so-net.ne.jp

Sincerely yours,

Yoshihisa Shimada, M.D., Ph.D.

---

## [Decision Letter · Decision Letter 3]

30 Dec 2020

PONE-D-20-26356R3

Small RNA sequencing to differentiate lung squamous cell carcinomas from metastatic lung tumors from head and neck cancers

PLOS ONE

Dear Dr. Shimada,

Thank you for submitting your manuscript to PLOS ONE. After careful consideration, we feel that it has merit but does not fully meet PLOS ONE’s publication criteria as it currently stands. Therefore, we invite you to submit a revised version of the manuscript that addresses the points raised during the review process.

We look forward to receiving your revised manuscript.

Kind regards,

Sumitra Deb, PhD

Academic Editor

PLOS ONE

Reviewers' comments:

Reviewer's Responses to Questions

**Comments to the Author**

1. If the authors have adequately addressed your comments raised in a previous round of review and you feel that this manuscript is now acceptable for publication, you may indicate that here to bypass the “Comments to the Author” section, enter your conflict of interest statement in the “Confidential to Editor” section, and submit your "Accept" recommendation.

Reviewer #3: (No Response)

2. Is the manuscript technically sound, and do the data support the conclusions?

Reviewer #3: Partly

3. Has the statistical analysis been performed appropriately and rigorously? 

Reviewer #3: No

4. Have the authors made all data underlying the findings in their manuscript fully available?

Reviewer #3: Yes

5. Is the manuscript presented in an intelligible fashion and written in standard English?

Reviewer #3: Yes

6. Review Comments to the Author

Reviewer #3: The authors need to provide statistical analysis for Fig 5b. Also since the cell lines used in Fig 5 are already established providing an image for cells as shown in 5a seems redundant. Instead authors may analyze more cell lines to strengthen their conclusion.

7. PLOS authors have the option to publish the peer review history of their article (what does this mean?). If published, this will include your full peer review and any attached files.

Reviewer #3: No

---

## [Author Response · Author response to Decision Letter 3]

27 Jan 2021

January 27, 2021

Joerg Heber

Editor-in-Chief

PLOS ONE 

Sumitra Deb

Academic editor 

PLOS ONE

Manuscript ID PONE-D-20-26356R3 entitled "Small RNA sequencing to differentiate lung squamous cell carcinomas from metastatic lung tumors from head and neck cancers"

Dear Dr. Heber, Deb

Thank you so much for your letter of 30-December-2020, stating your comments to our manuscript. We have carefully revised our manuscript and responded all the comments from reviewers as follows. 

Reviewer #3

The authors need to provide statistical analysis for Fig 5b. Also since the cell lines used in Fig 5 are already established providing an image for cells as shown in 5a seems redundant. Instead authors may analyze more cell lines to strengthen their conclusion.

Answer:

Thank you for your advice. We set up this in vitro analysis with several cell lines (3 lung cancers and 1 head and neck cancer) again. The miR3120 level in LSQCCs was significantly higher than that in the HNSQCC line, whereas the remaining gene levels were not significantly different between the 2 histological types. There is no consistency in the results by patient’s sample experiments and in vivo ones. Therefore, we exclude Figure 5 form our article. The following phrase is added. 

Change:

We checked the levels of the 4 miRNAs using extracellular vesicles derived from 4 LSQCC cell lines (EBA-1, H520, and LK-2 purchased from ATCC) and 1 HNSQCC cell line (BHY purchased from DSMZ) to examine whether the gene expressions in patients’ samples agree with those in cell lines. There were no significant differences in the expressions between two subtypes except for miR-3120 (data not shown). 

The comments offered by the reviewers and editors have been helpful in formulating what we believe is a stronger paper. We appreciate these thoughtful comments, and hope that our manuscript is now suitable for publication in PLOS ONE.

All related correspondence should be sent to Yoshihisa Shimada, M.D., Ph.D.

Department of Surgery, Tokyo Medical University Hospital

6-7-1 Nishishinjuku, Shinjyuku-ku, Tokyo, 160-0023, Japan

Phone: +81-(0)3-3342-6111, Fax: +81-(0)3-3342-6203

E-male: zenkyu@za3.so-net.ne.jp

Sincerely yours,

Yoshihisa Shimada, M.D., Ph.D.

---

## [Decision Letter · Decision Letter 4]

23 Feb 2021

Small RNA sequencing to differentiate lung squamous cell carcinomas from metastatic lung tumors from head and neck cancers

PONE-D-20-26356R4

Dear Dr. Shimada,

We’re pleased to inform you that your manuscript has been judged scientifically suitable for publication and will be formally accepted for publication once it meets all outstanding technical requirements.

Kind regards,

Sumitra Deb, PhD

Academic Editor

PLOS ONE

Additional Editor Comments (optional):

Reviewers' comments:

Reviewer's Responses to Questions

**Comments to the Author**

1. If the authors have adequately addressed your comments raised in a previous round of review and you feel that this manuscript is now acceptable for publication, you may indicate that here to bypass the “Comments to the Author” section, enter your conflict of interest statement in the “Confidential to Editor” section, and submit your "Accept" recommendation.

Reviewer #3: (No Response)

2. Is the manuscript technically sound, and do the data support the conclusions?

Reviewer #3: (No Response)

3. Has the statistical analysis been performed appropriately and rigorously? 

Reviewer #3: (No Response)

4. Have the authors made all data underlying the findings in their manuscript fully available?

Reviewer #3: (No Response)

5. Is the manuscript presented in an intelligible fashion and written in standard English?

Reviewer #3: (No Response)

6. Review Comments to the Author

Reviewer #3: (No Response)

7. PLOS authors have the option to publish the peer review history of their article (what does this mean?). If published, this will include your full peer review and any attached files.

Reviewer #3: No

---

## [Editor Report · Acceptance letter]

25 Feb 2021

PONE-D-20-26356R4 

Small RNA sequencing to differentiate lung squamous cell carcinomas from metastatic lung tumors from head and neck cancers 

Dear Dr. Shimada:

I'm pleased to inform you that your manuscript has been deemed suitable for publication in PLOS ONE. Congratulations! Your manuscript is now with our production department. 

Kind regards, 

on behalf of

Dr. Sumitra Deb 

Academic Editor

PLOS ONE